# Grammar-Forced Translation of Natural Language to Temporal Logic using LLMs

William English [1]   Dominic Simon [1]   Sumit Kumar Jha [2]   Rickard Ewetz [1]

## Abstract

Translating natural language (NL) into a formal language such as temporal logic (TL) is integral for human communication with robots and autonomous systems. State-of-the-art approaches decompose the task into a grounding of atomic propositions (APs) phase and a translation phase. However, existing methods struggle with accurate grounding, the existence of co-references, and learning from limited data. In this paper, we propose a framework for NL to TL translation called Grammar Forced Translation (GraFT). The framework is based on the observation that previous work solves both the grounding and translation steps by letting a language model iteratively predict tokens from its full vocabulary. In contrast, GraFT reduces the complexity of both tasks by restricting the set of valid output tokens from the full vocabulary to only a handful in each step. The solution space reduction is obtained by exploiting the unique properties of each problem. We also provide a theoretical justification for why the solution space reduction leads to more efficient learning. We evaluate the effectiveness of GraFT using the CW, GLTL, and Navi benchmarks. Compared with state-of-the-art translation approaches, it can be observed that GraFT improves the end-to-end translation accuracy by 5.49% and out-of-domain translation accuracy by 14.06% on average.

## 1. Introduction

Formal specifications play a crucial role in all systems that require autonomous reasoning, verification, or planning (Tellex et al., 2011; Abdulla et al., 2004). Ensuring that the behavior of such systems is bound to a set of known rules is essential to deploy them safely and reliably, especially when they are expected to operate without constant human supervision (Raman et al., 2013; Boteanu et al., 2016). Temporal Logics (TL) are a group of powerful formalisms that constitute the basis of most formal specification languages, allowing expressive description of the behavior of dynamic systems over time (Konur, 2013; Madsen et al., 2018). However, generating these specifications directly from natural language (NL) is an open problem, and achieving effective and efficient translation between NL and TL is an increasingly important task in advancing system automation (Chen et al., 2023; Fuggitti & Chakraborti, 2023; Cosler et al., 2023).

Early work on NL to TL translation focused on either on restricted natural language inputs (Raman et al., 2013) or template matching (Tellex et al., 2020). Although structured inputs are easy to map into TL, they place an undue burden on the user who may not have a technical background (Thistle & Wonham, 1986). On the other hand, template matching requires a substantial number of domain specific examples (Bombieri et al., 2023). More recently, NL to TL translation has been investigated using large language models (LLMs) (Cosler et al., 2023; Pan et al., 2023; Patel, 2019). This has involved approaches where LLMs are used to perform end-to-end translation. While such approaches can handle simple translations, the accuracy have been observed to be limited to 70% for more challenging test cases (Xu et al., 2024). More recent studies decompose the translation into a atomic propositions (APs) grounding phase and a translation phase. The two step approach is motivated by that domain specific terms will be extracted during the grounding phase, which facilitates domain agnostic translation (Fuggitti & Chakraborti, 2023). Existing works perform the grounding of APs using casual language model and few shot learning (Liu et al., 2023). The translation step is performed by a fine-tuned sequence-to-sequence model, where one token of the TL is predicted at the time (Chen et al., 2023). To boost the performance of the translation, studies have proposed to increase the size of the training data set using generative data augmentation (Chen et al., 2023). Unfortunately, it can be observed that ex-

[1]Department of Electrical and Computer Engineering, University of Florida, Gainesville, Florida [2]Knight Foundation School of Computing and Information Sciences, Florida International University, Miami, Florida. Correspondence to: William English <will.english@ufl.edu>.

*Proceedings of the $42^{nd}$ International Conference on Machine Learning*, Vancouver, Canada. PMLR 267, 2025. Copyright 2025 by the author(s).

isting AP grounding techniques struggle with consistently achieving high accuracy. In particular, for natural language inputs containing multiple references to the same AP, i.e., co-references. Moreover, the accuracy of the translation is highly dependent on the amount of available training data.

In this paper, we propose a framework for NL to TL translation called Grammar Forced Translation (GraFT). The framework is based on the observation that previous work solves both the grounding and translation steps by letting a language model iteratively predict the next token from the full token library. In contrast, GraFT reduces the complexity of both tasks by restricting the number of valid output token from the full library to only a handful in each step. The main contributions of this paper can be summarized, as follows:

1. The GraFT framework grounds the APs in the NL input using an masked language model (MLM). This both reduces the task complexity and restricts number of valid output tokens to the set of integers.

2. A fine-tuned sequence-to-sequence model is used to translate the grounded NL into TL. GraFT exploits the known grammar of the TL language, to place restrictions of the valid output tokens from the sequence-to-sequence model to a handful.

3. Mathematical justification are provided to explain why the restrictions on the output tokens lead to more efficient learning. We provide proofs for lower (or equal) cross-entropy as well as improved gradient alignment under grammar-forcing.

4. We evaluate the proposed framework on three natural language to temporal logic benchmarks- CW, GLTL, and Navigation. Compared with state-of-the-art approaches, we improve the accuracy of the end-to-end translation by 5.49% on average, by 14.06% on average in out-of-distribution tests.

The remainder of the paper is organized, as follows: Preliminaries are given in Section 2. The methodology is provided in Section 3 and experimental evaluation in Section 4. The paper is concluded in Section 5

## 2. Background

In this section, we review preliminaries on temporal logic, language modeling, and related work.

### 2.1. Temporal Logic

Temporal logic is a formal framework used to reason about propositions qualified in terms of time, allowing for the expression of statements about the temporal ordering of events (Konur, 2013). It extends classical propositional

logic by introducing modalities that capture time-related aspects, such as "always," "sometimes," "eventually," and "until." This paper deals with temporal logic but the concepts can easily be extended to signal temporal logic (Madsen et al., 2018) or linear temporal logic (Zhu, 2021). Temporal logic formulas are defined recursively, as follows:

$$\varphi ::= \pi^\mu |\neg\varphi|\varphi \wedge \psi|\varphi \vee \psi|\varphi \Rightarrow \psi| \bigcirc \varphi|\Diamond\varphi|\Box\varphi|\varphi \cup \psi,$$

where $\pi^\mu$ are the atomic predicates and both $\varphi$ and $\psi$ are temporal logic formula. $\neg$, $\wedge$, $\vee$, and $\Rightarrow$ are the logical operators negation, and, or, and implies, respectively. $\bigcirc$, $\Diamond$, $\Box$, $\cup$, are the temporal operators *next*, *eventually*, *always*, and *until* respectively.

A sample NL to TL translation problem would involve translating the natural language sentence "Go to the red room and push the box into the green room." into "$\Diamond(red\_room \wedge \Diamond green\_room)$".

### 2.2. Language Modeling

In this section, we review popular language modeling approaches such as masked language modeling, sequence-to-sequence modeling, and causal language modeling.

**Masked Language Modeling (MLM):** Masked language modeling is a training approach used primarily in NLP where certain words in a sentence are randomly masked or hidden, and the model is tasked with predicting these missing words based on the surrounding context (Devlin et al., 2019). This technique helps the model understand relationships between words and enhances its ability to generate coherent and contextually relevant text. MLM is the foundation for training of models like BERT (Devlin et al., 2019), DistilBERT (Sanh et al., 2020), and ROBERTA (Liu et al., 2019). MLMs have demonstrated excellent performance of tasks with bidirectional representations.

**Sequence-to-Sequence Language Modeling (Seq2Seq):** Sequence to sequence modeling is a framework designed for tasks where input and output are both sequences, such as translation or summarization (Raffel et al., 2020). In this architecture, two neural networks, typically an encoder and a decoder, work in tandem: the encoder processes the input sequence and compresses it into a fixed-size context vector, while the decoder generates the output sequence from this representation. Seq2Seq has been used to train models such as T5 (Raffel et al., 2020), Bart (Lewis et al., 2019), and Pegasus (Zhang et al., 2020). The Seq2Seq approach works well for tasks that require understanding the entire input context before generating the output, enabling applications like machine translation, chat-bots, and text summarization.

**Causal Language Modeling (CLM)** Causal language modeling, often associated with autoregressive models, focuses on predicting the next word in a sequence given the previ-

| Approach | AP grounding | | TL Translation | | |
| --- | --- | --- | --- | --- | --- |
| | Model | Vocab Size | Model | Decoding Strategy | Vocab Size |
| NL2LTL (Fuggitti & Chakraborti, 2023) | '-' | - | CLM | - | 100,256 |
| NL2Spec (Cosler et al., 2023) | '-' | - | CLM | - | 50,257 |
| NL2TL (Chen et al., 2023) | CLM | 100,256 | Seq2Seq | - | 32,182 |
| GraFT (proposed) | MLM | 6 | Seq2Seq | Grammar-Constrained | $1 \leq |V| \leq 20$ |

Table 1: Contrast between the use of LLMs within the proposed translation and state-of-the-art approaches. A '-' denotes that the step was not performed. GraFTs MLM vocab size is 6 due to the 6 labels relating to APs (0 for non-AP or 1-5), and between 1 and 20 for the translation- the minimum and maximum number of valid tokens at any given state.

ous words (Achiam et al., 2024). This method is inherently directional and processes the input in a left-to-right manner. Models that are trained using CLM include the prominent GPT family (Achiam et al., 2024). This approach is effective for tasks that require coherent prompt-based text generation.

### 2.3. Related Work

In this section, we provide an overview of recent attempts at translating natural language into formal language specifications and temporal logic using LLMs. The translation can be decomposed into two main steps: 1) grounding of atomic predicates and 2) NL to TL translation.

**AP Grounding:** Atomic predicates (APs) are state variables or events that temporal logic reason over. While the logical statements are the same for different domains, the definition of the atomic predicates can vary substantial between different applications. For example, the APs used to describe the operation of a traffic light are very different from the APs used to describe the operation of an autonomous robot. Therefore, it was proposed the atomic predicates within the NL should be extracted (or grounded) in a pre-processing step before the translation into TL (Chen et al., 2023; Liu et al., 2023; Hsiung et al., 2021). The grounded NL would be more similar across domains and allow cross-domain adaptation with the use of less training data. This was performed by replacing each atomic predicate with a $prop\_x$ variable in (Chen et al., 2023), descriptive single words in (Liu et al., 2023), and templates in (Hsiung et al., 2021). These grounding approaches improve downstream translations but can introduce errors, including hallucinated keywords and difficulties handling co-references, where different words refer to the same AP.

**NL to TL Translation:** The translation of natural language (or grounded natural language) using LLMs has been explored in (Fuggitti & Chakraborti, 2023; Cosler et al., 2023; Chen et al., 2023). The most straightforward approach to NL to TL translations is to used an large off-the-shelf CLM (Chen et al., 2023). The accuracy of the approach can be enhanced if restrictions are placed on the NL descriptions and by providing examples with few shot prompting. Breaking down the problem into sentences and translat-

ing each of them independently was proposed in (Cosler et al., 2023). Several works have investigated fine-tuning sequence-to-sequence models to perform translation (Pan et al., 2023; Patel, 2019; Chen et al., 2023). The performance improvements achieved through fine-tuning are often correlated with the amount of available training data. Data augmentation to generate training data using LLMs was proposed in (Pan et al., 2023). An alternative to data dependency is deploying the translation model in a reinforcement learning environment, using rewards for fine-tuning. However, these methods overlook TL's structured, context-free grammar.

**Proposed Method:** We observe that previous approaches to grounding and translation use language models without exploiting the unique properties of the problems to enhance learning. For both tasks, a language model is used to predict a token from the models full token library. In the proposed GraFT framework, we propose to reduce the complexity of both tasks by placing restrictions on the valid output tokens. In particular, the grounding is reformulated into a masked language modeling problem where the output tokens are restricted to integers. For the translation, the temporal logic grammar is used to dynamically reduce the number of valid output tokens to a handful. The main differences compared with previous work are shown in Table 1.

### 3. Methodology

In this section, we present the methodology of the GraFT framework that converts natural language into temporal logic. The framework consists of two steps: grounding of atomic predicates (APs) and grounded NL to TL translation. The first step uses an MLM (BERT (Devlin et al., 2019)) to identify the atomic predicates, which is detailed in Section 3.1. The second step uses a Seq-2-Seq model (T5 (Raffel et al., 2020)) to translate the lifted natural language given in the previous step into lifted linear temporal logic, as outlined in Section 3.2. Finally, the lifted APs from step one are inserted into the lifted LTL, yielding the final translation result. An overview of the flow of the framework is shown in Figure 1.

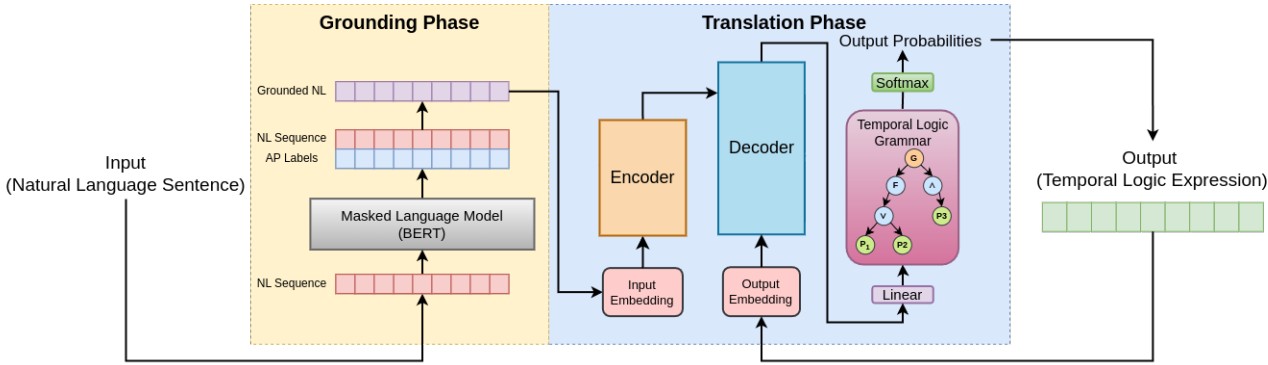

Figure 1: An overview of the GraFT framework of end-to-end translation of NL to TL.

### 3.1. Grounding of Atomic Predicates

In this section, we describe how the GraFT framework extracts the atomic predicates from the NL input. AP grounding is the process of substituting the atomic predicates (APs) within the NL input to obtain *Grounded NL*. Before we outline our approach, we examine the characteristics of performing the AP grounding using different types of language modeling.

**Failure Analysis of CLMs:** We compare performing the AP grounding using a CLM (GPT-4o) and a MLM (BERT) in Figure 2. We first focus on performing the AP extraction using a CLM as in (Chen et al., 2023). Causal language models are trained to perform next-word prediction rather than to predict masked tokens in a sequence. While CLMs are regarded for their reasoning abilities, and are capable of extracting the APs as shown at the top of the figure, the sequence returned by the model is not 1-to-1 with the input sequence. While the CLMs may "understand" that it is supposed to replicate the input sentence with the APs masked, it is prone to introducing errors by modifying the sentence

to sound more natural. For example, it is not surprising to observe that the causal model has slightly modified the input by dropping the word "*eventually*", which introduces an error in the subsequent translation, as it is a key word for temporal logic. Moreover, the output of the CLM must be parsed as it is not guaranteed to follow the format shown in the few shot prompting examples. Conversely, masked language models (MLMs) are trained to predict labels on the input tokens, which we hypothesize is much more effective solution for solving this task. In the bottom of the figure, it can be observed that our proposed grounding approach only assigns an indicator variable to each of the tokens in the input, i.e., the indicator variable determines if a token is part of an AP or not.

**Proposed Grounding using MLM:** The AP grounding in the GraFT framework is performed by fine-tuning a MLM. Using the fine-tuning process, we teach the MLM to assign an indicator variable $I_i$ to each token $i$ of the input. The indicator variables $I$ form a list of integers. In integer AP grounding, each AP in the input string is assigned an integer ID, and tokens which are part of a reference to that AP

```
Causal          Input NL:              "Redirect the asset near 24th avenue. Follow if it leaves there."
Language
Model           Grounded NL:           "[prop_1] [prop_2] [prop_3]. [prop_4] if [prop_2] [prop_5] [prop_3]."
(GPT)
-·-·-·-·-·-·-·-·-·-·-·-·-·-·-·-·-·-·-·-·-·-·-·-·-·-·-·-·-·-·-·-·-·-·-·-·-·-·-·-·-·-·-·-·
Masked          Input NL:              "Redirect the asset near 24th avenue. Follow if it leaves there."
Language
Model           Integer Grounding:  [1,     0,   2,   0,   3,   3,   0,   4,   0,   2,   5,   3, 0]
(BERT)
                Grounded NL:         "[prop_1] the [prop_2] near [prop_3]. [prop_4] if [prop_2] [prop_5] [prop_3]."

                Integer AP Mapping: [prop_1] = "Redirect"
                                    [prop_2] = "asset", "it"
                                    [prop_3] = "24th avenue", "there"
                                    [prop_4] = "Follow"
                                    [prop_5] = "leaves
```

Figure 2: AP grounding using LLMs trained using different types of language modeling.

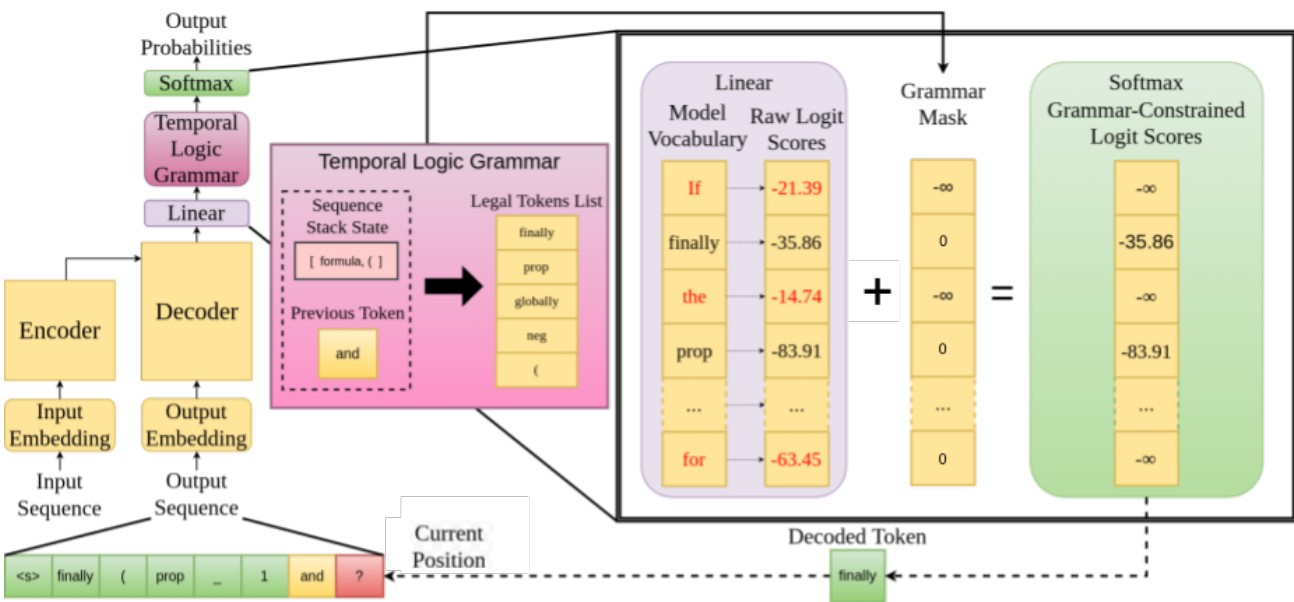

Figure 3: At each training step, prior to performing `SoftMax` on the output logits in preparation for computing $L$, we observe the label at position $t$ in the target sequence $L$. Given this token and the current state of the TL parsing stack, we can obtain a set of valid token that may proceed that label. For each output logit $Z_t$, we set the score for all tokens outside of $V_t$ to $-\infty$.

are labeled with the respective ID. An example of this AP grounding approach is given in Figure 2.

$$I_i = \begin{cases} 0, & I_n \text{ is not part of an AP,} \\ n, & I_n \text{ is part of the } n_{\text{th}} \text{ AP.} \end{cases} \quad (1)$$

The objective of the fine-tuning is to predict the indicator variables $I$. We use standard cross-entropy loss for the training using an annotated portion of the Navigation dataset (Wang et al., 2021). Notably, each of the MLMs trained only on the Navigation dataset demonstrate nearly perfect performance for other datasets. These results are presented in Table 3.

### 3.2. Grounded NL to TL Translation

In this section, we describe how the grounded NL is translated into TL. The translation is performed using a Seq2Seq model. The key idea is to impose the grammar of the TL when decoding the output from the model. In particular, we introduce two grammar-based augmentations: a grammar-forcing strategy during training, and grammar-constrained decoding during inference. The details are provided in Section 3.2.1. Lastly, the section is concluded with a theoretical basis for the improved efficiency in learning. The details of the theoretical motivation are provided in Section 3.2.2. An overview of the proposed translation framework is shown in Figure 3.

#### 3.2.1. GRAMMAR CONSTRAINED DECODING

In order to exploit the grammatical structure of the target language, we construct a logits processor that ensures the sequence being decoded obeys the rules. Because of this, we can guarantee that sequences produced by GraFT are grammatically correct temporal logic expressions. Our implementation of the grammar-constrained logits processor is described in Algorithm 1.

Algorithm 1 effectively adjusts the logit scores returned by sequence-to-sequence models to decode logits during generation. The first token in the sequence is restricted to the known list of valid initial tokens, and so for that logit we apply a mask that erases all tokens that can not follow from the initial state of the grammar. For all subsequent tokens, we use the temporal logic grammar to determine which tokens may proceed the previously decoded token, as well as which tokens may occur given the current state of the temporal logic grammar. We now turn to further explanation of this training procedure.

As previously stated, we apply grammar constrained decoding during training. However, rather than pass the previous token *generated by the model*, we pass the previous token *in the target sequence*. This technique is reminiscent of teacher-forcing (Hao et al., 2022), as we use ground truth labels to inform the model of how it may generate sequences during training. It is similar to the grammar-constrained decoding we use during inference, except we use the tokens of

the target sequence to update the grammar state and retrieve valid tokens. We now turn to a theoretical justification for this technique.

### 3.2.2. THEORETICAL BASIS FOR GRAMMAR-FORCING

We know intuitively that by zeroing-out known invalid tokens from the output logits vector, we can reduce our cross-entropy loss. The following is a more formal argument for this claim. Firstly, we introduce our notation and propose that our approach provides some guaranteed improvements. We then explain how we reach this conclusion in formal terms. We provide additional justification in Section A.4 of the Appendix.

**Preliminaries:** Let $\mathcal{V}$ be the vocabulary of the model $p(\theta)$. At each decoding step $t$, the model returns a vector of logits $Z = (z_1, z_2, ..., z_n)$ where $z_n = (v_1, v_2, ...v_{|\mathcal{V}|}) \in \mathbb{R}^{|\mathcal{V}|}$. The standard softmax distribution over a single logit is

$$p(v) = \frac{\exp(z_v)}{\sum_{u \in \mathcal{V}} \exp(z_u)}, \qquad v \in \mathcal{V}$$

if the label at step $t$ is $y \in \mathcal{V}$, the cross-entropy loss is

$$L(z, y) = -\log\left(\frac{\exp(z_y)}{\sum_{u \in \mathcal{V}} \exp(z_u)}\right)$$

Now suppose that at step $t$, only a subset $\mathcal{V}_t \subseteq \mathcal{V}$ contains *grammatically valid* tokens. We *mask out* all invalid tokens by setting their logits to $-\infty$. Define the transformed logits

$$z'_v = \begin{cases} z_v, & \text{if } v \in \mathcal{V}_t, \\ -\infty, & \text{if } v \notin \mathcal{V}_t, \end{cases}$$

and the corresponding distribution

$$p'(v) = \begin{cases} \dfrac{\exp(z_v)}{\sum_{u \in \mathcal{V}_t} \exp(z_u)}, & \text{if } v \in \mathcal{V}_t, \\ 0, & \text{if } v \notin \mathcal{V}_t, \end{cases} \qquad (2)$$

The **grammar-forced cross-entropy loss** becomes

$$L'(\mathbf{z}, y) = -\log p'(y) = -\log\left(\frac{\exp(z_y)}{\sum_{u \in \mathcal{V}_t} \exp(z_u)}\right),$$

assuming $y \in \mathcal{V}_t$.

**Masking Invalid Tokens Improves Optimization** Let $y \in \mathcal{V}_t$ be the target , and let $\mathbf{z}$ be the logit vector in that position at some iteration of training. Then the grammar-forced cross-entropy $L'(\mathbf{z}, y)$ is never larger than the standard cross-entropy $L(\mathbf{z}, y)$. Furthermore, the gradient of $L'$ focuses updates only on valid tokens, thereby reducing the effective search space and often yielding faster or more stable convergence.

**(1) Lower (or Equal) Cross-Entropy.**

By construction,

$$L(\mathbf{z}, y) = -\log\left(\frac{\exp(z_y)}{\sum_{v \in \mathcal{V}} \exp(z_v)}\right)$$

$$L'(\mathbf{z}, y) = -\log\left(\frac{\exp(z_y)}{\sum_{u \in \mathcal{V}_t} \exp(z_u)}\right).$$

Since $\mathcal{V}_t \subseteq \mathcal{V}$, we have

$$\sum_{v \in \mathcal{V}_t} \exp(z_v) \leq \sum_{v \in \mathcal{V}} \exp(z_v).$$

Thus

$$\frac{\exp(z_y)}{\sum_{u \in \mathcal{V}_t} \exp(z_u)} \geq \frac{\exp(z_y)}{\sum_{v \in \mathcal{V}} \exp(z_v)},$$

which implies

$$-\log\left(\frac{\exp(z_y)}{\sum_{u \in \mathcal{V}_t} \exp(z_u)}\right) \leq -\log\left(\frac{\exp(z_y)}{\sum_{v \in \mathcal{V}} \exp(z_v)}\right).$$

Hence $L'(\mathbf{z}, y) \leq L(\mathbf{z}, y)$. Equality can occur if $z_v = -\infty$ for all $v \notin \mathcal{V}_t$, which is precisely the masking scenario.

**(2) More Focused Gradient (Better Alignment).**

For standard cross-entropy, the gradient with respect to each logit $z_k$ is:

$$\frac{\partial L(\mathbf{z}, y)}{\partial z_k} = p(k) - \mathbf{1}[k = y],$$

where $p(k) = \exp(z_k) / \sum_{v \in \mathcal{V}} \exp(z_v)$.

Under grammar-forcing,

$$\frac{\partial L'(\mathbf{z}, y)}{\partial z_k} = p'(k) - \mathbf{1}[k = y],$$

We also recall the probability distribution under grammar forcing 2 and observe that

$$\frac{\partial L'(\mathbf{z}, y)}{\partial z_k} = 0, \quad \forall \ k \notin \mathcal{V}_t.$$

Recalling that $y \in \mathcal{V}_t$, we observe that no gradient signal is *wasted* on tokens that can *never* be correct, allowing each update step to invest more effective capacity into discriminating among valid tokens.

| Approach | Data Quantity | AP Grounding Model | Translation Model | CW (%) | GLTL (%) | Navi (%) |
|---|---|---|---|---|---|---|
| NL2LTL (Fuggitti & Chakraborti, 2023) | - | None | GPT-4o-mini | 69.90 | 74.40 | 65.30 |
| NL2Spec (Cosler et al., 2023) | - | None | GPT-4o-mini | 78.60 | 68.40 | 71.60 |
| Ungrounded Seq2Seq | 500 | - | T5 | 59.60 | 46.80 | 43.40 |
| NL2TL (Chen et al., 2023) | 500 | GPT-4o-mini | T5 | 93.00 | 83.80 | 80.40 |
| NL2TL (Chen et al., 2023) | 500 | GPT-4 | T5 | 91.50 | 82.60 | 83.70 |
| **GraFT (proposed)** | 500 | BERT | T5 | **97.70** | **91.50** | **85.00** |
| Ungrounded Seq2Seq | 2000 | - | T5 | 68.10 | 55.90 | 56.30 |
| NL2TL (Chen et al., 2023) | 2000 | GPT-4o-mini | T5 | 98.20 | 97.40 | 86.70 |
| NL2TL (Chen et al., 2023) | 2000 | GPT-4 | T5 | 96.70 | 96.20 | 90.00 |
| **GraFT (proposed)** | 2000 | BERT | T5 | **99.90** | **99.80** | **99.10** |

Table 2: Performance comparison of different models for end-to-end translation with training data from each dataset. The BERT model used for AP Grounding is trained on the same data used to train the translation model.

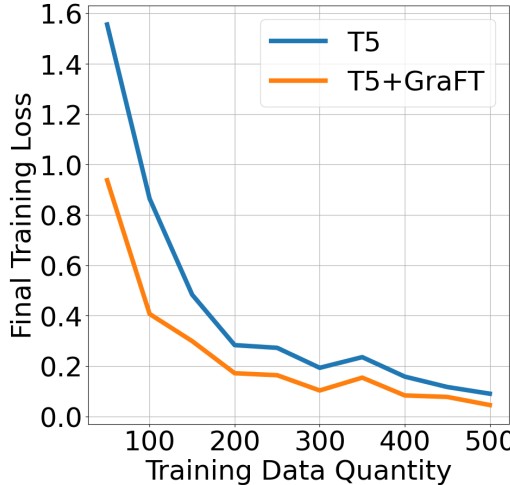

Figure 4: A comparison of training loss for T5 with vs without grammar-forcing during training.

## 4. Experimental Evaluation

We conducted our evaluation on a machine with one NVIDIA RTX 4070 Ti Super GPU, one Intel i9-14900KF 32 Core CPU, and 64GB of RAM. Our evaluation datasets include Navigation (Wang et al., 2021), GLTL (Gopalan et al., 2018), and CW (MacGlashan et al., 2015). Some statistics on these datasets are given in the appendix A.1. We evaluate performing the AP grounding using the MLMs BERT, RoBERTa, and DistilBERT. Each AP grounding model was trained for 3 epochs at a learning rate of 1e-5. We also evaluate performing the grounded NL to TL translation. Each translation model was trained for 3 epochs at a learning rate of 2e-5. We perform our evaluation of the translation models and end-to-end approaches using 1000 examples from each dataset. We first perform ablation studies to evaluate the impact of each of the different parts of the GraFT

framework in Section 4.1. Next, we provide end-to-end translation results in Section 4.3.

### 4.1. Ablation Studies of GraFT

| Model | Objective | CW (%) | GLTL (%) | Navi (%) |
|---|---|---|---|---|
| GPT-4o-mini | Causal | 97.76 | 95.84 | 83.97 |
| GPT-4o | Causal | 95.02 | 93.53 | 86.08. |
| GPT-4 | Causal | 96.24 | 94.68 | 87.28 |
| DistilBERT | Masked | 95.80 | 93.83 | 99.99 |
| RoBERTa | Masked | 98.34 | 96.96 | 99.99 |
| BERT | Masked | **98.58** | **97.35** | 99.99 |

Table 3: The table evaluates the models' accuracy on the AP Masking task using 1000 examples from the Navi dataset.

The effectiveness of performing AP extraction using an MLM compared with a CLM is evaluated in Table 3. For the AP Grounding evaluation, we use top-1 accuracy (fail/succeed) for each test. We use 1000 unseen examples from each dataset. We observe that GPT-4o-mini performs reasonably well on both CW and GLTL, but struggles with identifying APs from the Navi dataset. The performance of each of the BERT models is quite impressive across each domain. All three models both achieve almost 100% accuracy on unseen in-domain examples, with BERT and RoBERTa maintaining high performance on the out-of-domain datasets as well.

### 4.2. Grounded Translation Results

In Figure 5, we compare the accuracy of three T5 models with respect to training data quantity. We use the T5 checkpoint provided at the HuggingFace-hosted repository (Raffel et al., 2020) as the base for each model. We then trained two versions of this model in accordance with the two frame-

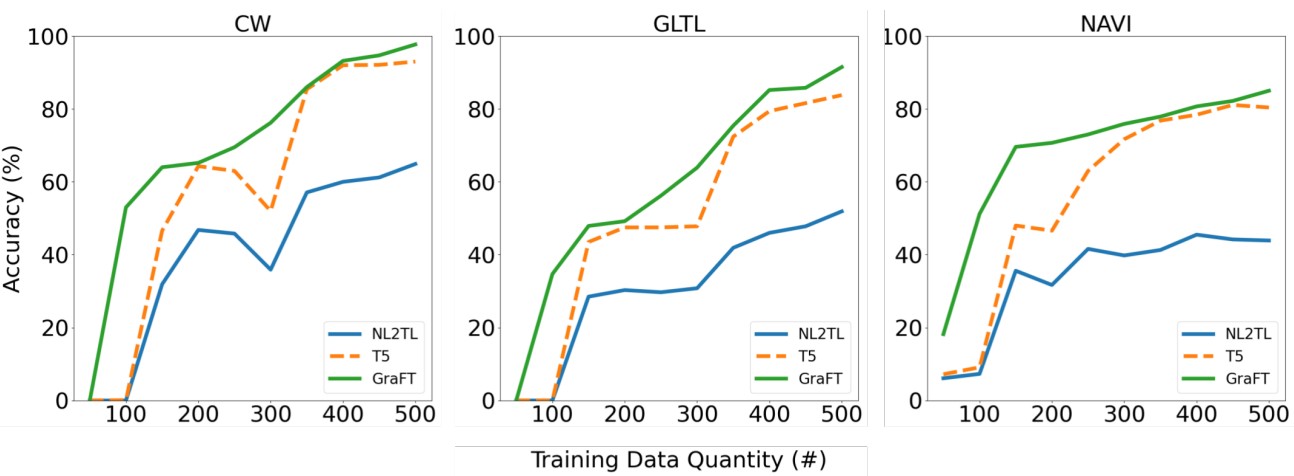

Figure 5: Accuracy results of three T5-based translation models. The T5 framework is a baseline evaluated with ground-truth lifted NL specifications as input. GraFT and NL2TL are evaluated using their respective lifting approaches. Each translation model is trained with LR= 2e-5 for 3 training epochs.

| Approach | Training Data | CW (%) | GLTL (%) | Navi (%) |
|---|---|---|---|---|
| NL2TL (Chen et al., 2023) | GLTL+Navi | 60.2 | 44.0 | 55.3 |
| **GraFT (proposed)** | GLTL+Navi | **63.2** | **48.3** | **73.0** |
| NL2TL (Chen et al., 2023) | CW+Navi | 21.2 | 23.1 | 51.6 |
| **GraFT (proposed)** | CW+Navi | **58.8** | **41.1** | **73.1** |
| NL2TL (Chen et al., 2023) | CW+GLTL | 63.0 | 45.3 | 14.6 |
| **GraFT (proposed)** | CW+GLTL | **70.0** | **56.2** | **21.2** |

Table 4: Performance comparison of different models for end-to-end translation. Here, fine-tuning is performed with respect to two of the datasets, and evaluation is performed using all three datasets.

works using their respective grounded natural language and temporal logic pairs. The NL2TL and T5 both use standard cross-entropy loss, which we compare against our grammar-forcing method described in 3.2. The baseline T5 model uses the ground-truth lifted natural language, contributing to its high accuracy. Our first observation as that our approach yields greater accuracy at each training data quantity, most notably with smaller quantities of training data. Our second observation is that the GraFT model never suffers a reduction in accuracy as a result of additional training data, while the control model suffers heavily, particularly in the CW evaluation. We believe that less gradient noise during training allows GraFT to continue learning with the introduction of unseen training data, while the control model may become confused when exposed to new sequence pairs. We observe that grammar-forcing improves accuracy by 0.9% - 42.10%.

### 4.3. End-to-end results

We present the end-to-end results of GraFT and other temporal logic (TL) translation approaches in Table 2. All methods

are trained on datasets containing examples from each domain, and we specify the models used for translating natural language (NL) to TL and for identifying atomic propositions (APs) where applicable. GraFT outperforms all compared approaches, achieving an average accuracy of 99.6% across the datasets. NL2LTL and NL2Spec under-perform GraFT by at least 12.8% to 21.24% on CW, GLTL, and Navigation datasets, respectively. These approaches do not perform AP grounding and rely solely on CLMs for translation. GraFT demonstrates ∼5% improvement over NL2TL when the data quantity is 500. This gap closes slightly when the data quantity is 2000, excepting the Navi dataset, where GraFT demonstrates an improvement of 12.40%. We attribute GraFT's superior performance to its training approach, which enhances domain transferability over standard T5 models fine-tuned with cross-entropy loss.

Additionally, NL2TL's use of GPT for AP grounding underperforms on the Navi dataset. We evaluate performance on out-of-distribution data in Table 4. GraFT achieves higher accuracy on all out-of-distribution tests, demonstrating av-

erage improvements by 8.33% on GLTL+NAVI, 25.7% on CW+NAVI, and 8.16% on CW+GLTL. The notable benefit observed in the CW+NAVI evaluation is NL2TL's failure to learn the CW distribution despite its inclusion in the training set. On the other hand, GraFT was able to learn this distribution more successfully. This shows the potential for generalization if a modest pre-training dataset for NL to TL translation is available in the target domain.

## 5. Conclusion

In this paper, we present GraFT, a framework for natural language to temporal logic translation. We demonstrate that grammar-forced training improves generalization and reduces domain-specific training data requirements. We also apply MLMs to more cheaply and accurately perform AP grounding prior to translation. The combination of a problem-specific training approach and AP grounding with BERT results in a more robust and generalized natural language to temporal logic translation framework that preserves NL AP segments from the original input to provide the NL-AP mapping required to interpret the TL expression. Future work in this area will include collecting and synthesizing diverse NL-TL datasets to further evaluate the transferability of translation models.

## Acknowledgements

This material is in part sponsored by UF startup funds and DARPA under agreement number FA8750-23-2-0501. The views and conclusions contained herein are those of the authors and should not be interpreted as necessarily representing the official policies or endorsements, either expressed or implied, of DARPA or the U.S. Government.

## Impact Statement

This paper presents work whose goal is to advance the field of Machine Learning. There are many potential societal consequences of our work, none which we feel must be specifically highlighted here.

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

# A. Appendix

In this appendix, we provide supplementary results, analysis, and justification for our approach. We first provide quantitative information about the NL-LTL corpora used in our experiments in Section A.1. In Section A.2, we provide our LTL logits processing algorithm and discuss its usage during training and inference. In Section A.3 we provide an ablation of lifting models with various AP quantities. Lastly, in Section A.4, we provide extended theoretical support for our claims made in Section 3.2.2

## A.1. Corpus Statistics

In this section, we present diversity metrics for each corpora used in our experiments. Each corpus is roughly comparable in terms of linguistic diversity. We also note that the CW corpus contains significantly fewer unique grounded LTL expressions.

| Domain | # NL | # LTL | # Vocab |
|---|---|---|---|
| Navi (Wang et al., 2021) | 7,079 | 142 | 139 |
| GLTL (Gopalan et al., 2018) | 11,153 | 188 | 203 |
| CW (MacGlashan et al., 2015) | 2,130 | 39 | 196 |

Table 5: Statistics of NL-TL corpora used in our training and evaluation. The **# NL** column gives the number of unique NL input sentences, the **# LTL** column gives the number of unique TL expressions, and **# Vocab** gives the total number of unique words.

## A.2. Algorithm 1: LTL Logits Processor

In this section, we present the algorithm used for our grammar-forced translation approach. During translation, the state variable tracks the current state of an LTL parser. As the logits are decoded by the model, we zero-out invalid token predictions, almost identical to our approach used in grammar-forced training, shown in 3. The only difference is that during training, we use the ground-truth token labels to maintain the state of the translation. During inference, we instead maintain the state based on the model's own outputs.

---

**Algorithm 1** Temporal Logic Logits Processor

**Input:** Input IDs $I$, Scores $S$
state = grammar.init_state()
**for** $i$ in range($I$) **do**
  **if** i $>$0 **then**
    last_token = $I_{i-1}$
    state.update(last_token)
  **end if**
  legal_tokens = grammar.get_valid(state)
  mask = [1 in range($S$)]
  mask[legal_tokens] = 0
  S[$i$, mask] = $-\infty$
**end for**
**Output:** Scores $S$

---

### A.3. Lifting Ablations

We evaluate four off the best-performing lifting models on lifting sequences with more than 5 atomic predicates. To obtain these sequences, we concatenated existing entries that did not share APs and re-labeled the newly conjoined AP dictionaries. The results demonstrate promising scalability in MLM-based approaches, while the LLMs performance suffers rapid decline as AP quantities increase.

| Model | Objective | CW (%) | GLTL (%) | Navi (%) |
|---|---|---|---|---|
| GPT-4 | Causal | 81.88 | 80.41 | 83.02 |
| DistilBERT | Masked | 94.20 | 91.75 | 99.99 |
| RoBERTa | Masked | 96.37 | 95.66 | 99.99 |
| BERT | Masked | 97.10 | 96.52 | 99.99 |

Table 6: AP grounding results for 6–10 APs (Range B).

| Model | Objective | CW (%) | GLTL (%) | Navi (%) |
|---|---|---|---|---|
| GPT-4 | Causal | 70.34 | 69.80 | 72.24 |
| DistilBERT | Masked | 92.86 | 90.63 | 98.54 |
| RoBERTa | Masked | 95.13 | 94.67 | 99.73 |
| BERT | Masked | 95.78 | 96.44 | 99.91 |

Table 7: AP grounding results for 11–15 APs (Range C).

### A.4. Extended Theoretical Basis for Grammar-constrained Optimization

Here, we outline an argument for our claim that imposing grammar constraints (that are not violated by the distribution of training data) during training can improve the convergence rate of stochastic gradient descent (SGD). Concretely, Theorem A.4 shows that there is a *variance term $M$* that appears in the rate. We proceed by showing that if applies a mask over token sequences that necessarily do not appear the distribution, this variance term decreases to $M_{(g)} < M$, hence reducing the constant factor in the $O(\frac{1}{\sqrt{N}})$ bound, indicating *faster convergence in practice*, particularly when $N$ is small. This section relies heavily on Theorem A.4, and we recommend reviewing section 4.3 of (Bottou et al., 2016) for a detailed proof of convergence bounds on SGD in the context of a non-convex objective function with non-diminishing stepsize.

**Claim A.4** : Let $\ell(\theta; x)$ and $\ell^{(g)}(\theta; x)$ be the cross-entropy loss of an unconstrained and constrained language model, respectively. Suppose both losses are differentiable in $\theta$ and have bounded gradients.

$$\mathbb{E}[||\nabla \ell^{(g)}(\theta_n)||^2] \leq \mathbb{E}[||\nabla \ell(\theta_n)||^2]$$

**Theorem: SGD Convergence Bounds (Nonconvex, Diminishing Stepsize) (Bottou et al., 2016)** :

$$\sum_{n=1}^{N} a_n \mathbb{E}\left[||\nabla \ell(\theta_n)||_2^2\right] \leq \frac{2(\mathbb{E}[\ell(\theta_1)] - \ell_{inf})}{\mu} + \frac{LM}{\mu} \sum_{n=1}^{N} a_k^2$$
$$\approx O\left(\frac{M}{\sqrt{N}}\right),$$

where - $L$ is a smoothness constant, - $\mu > 0$ is a parameter related to descent conditions, - $\bar{a} > 0$ is a stepsize parameter, and - $M$ is a uniform bound on the second moment (or variance) of the stochastic gradients (see Assumption A.4).

**Definition 1 (Unconstrained Language Model):**

$$p_\theta(x_t|x_{<t}) \qquad\qquad = \texttt{SoftMax}(z_\theta(x_{<t}))[x_t], \qquad\qquad (3)$$

where $z_\theta(x_t) \in \mathbb{R}^{|V|}$ is the logit vector for all tokens in the full vocabulary $V$.

**Definition 2 (Grammar-Constrained Language Model):**

$$\mathcal{M}_t := \begin{cases} 1 & \text{if } v \in V_t, \\ -\infty & \text{otherwise.} \end{cases} \tag{4}$$

$$p_\theta^{(g)}(x_t | x_{<t}) = \text{SoftMax}(z_\theta(x_{<t}) \odot \mathcal{M}_t)[x_t], \tag{5}$$

**Definition 3 (Cross-Entropy Loss at Iteration $n$)**  Let $\{x_1, x_2, \ldots, x_T\}$ be a sequence drawn from a data distribution $D$. At iteration $n$, the model parameters are $\theta_n$. Then we define the cross-entropy loss for a single sequence $x$ as

$$\ell(\theta_n; x) = -\sum_{t=1}^{T} \log p_{\theta_n}(x_t | x_{<t}).$$

When averaging over all sequences $x$ in the data distribution $D$, we obtain the *expected cross-entropy loss* at iteration $n$:

$$L(\theta_n) = \mathbb{E}_{x \sim D}[\ell(\theta_n; x)] = -\mathbb{E}_{x \sim D}\left[\sum_{t=1}^{T} \log p_{\theta_n}(x_t | x_{<t})\right].$$

**Assumption 1 (Correct Grammar)**  If $x_t$ occurs in the data for prefix $x_{<t}$, then $x_t \in V_t \subseteq V$. Therefore, grammar-masking does not affect tokens that actually appear.

**Assumption 2 (Bounded Variance of SG Estimation)**  :  There exist constants $m, m_v \geq 0$ such that

$$\mathbb{V}[g(\theta, \varepsilon)] \leq m + m_v ||\nabla \ell(\theta)||_2^2, \text{where } x \approx D, \tag{6}$$

**Proof of Claim A.4**  : Consider a parameter vector $\theta$ and a random sample $x \sim D$. For the unconstrained model (3), we observe

$$0 < \mathbb{E}\left[\frac{\partial \nabla \ell(\theta, x)}{z_\theta(x_{<t})[v]}\right] \forall v \in V$$

by contrast, in the grammar-constrained model (4), we can observe

$$\mathbb{E}\left[\frac{\partial \nabla \ell^{(g)}(\theta, x)}{z_\theta(x_{<t})[v]}\right] = 0 \forall v \notin V_t.$$

Hence, coordinate-wise,

$$\left|\nabla \ell^{(g)}(\theta; x)[v]\right| \leq \left|\nabla \ell(\theta; x)[v]\right| \quad \forall v.$$

Therefore,

$$||\nabla \ell^{(g)}(\theta; x)||^2 \leq ||\nabla \ell(\theta; x)||^2$$

Taking expectation over $x \sim D$ and any internal sampling noise, we obtain

$$\mathbb{E}[||\nabla \ell^{(g)}(\theta)||^2] \leq \mathbb{E}[||\nabla \ell(\theta)||^2].$$

Defining

$$M_{(g)} := \sup_\theta \sqrt{\mathbb{E}[||\nabla \ell^{(g)}(\theta)||^2]},$$

$$M := \sup_\theta \sqrt{\mathbb{E}[||\nabla \ell(\theta)||^2]},$$

we see $M_{(g)} \leq M$. Strict inequality $M_{(g)} < M$ arises whenever there *is* at least one token $v$ that the grammar forbids but the unconstrained model might otherwise assign non-negligible probability to. Hence the grammar-constrained model has a *strictly smaller* variance constant in the sense used in Theorem A.4. By Theorem A.4, the asymptotic SGD bound in the nonconvex/diminishing-stepsize scenario is on the order of $O(M/\sqrt{N})$. Since $M_{(g)} < M$, replacing $M$ by $M_{(g)}$ yields *tighter* constants in the finite-time and asymptotic terms. Thus, grammar-constrained optimization *accelerates* convergence by reducing stochastic gradient variance.

