# OpenReview forum: "Grammar-Forced Translation of Natural Language to Temporal Logic using LLMs"
_ICML.cc/2025/Conference — ICML 2025 poster_

### Official Review · Reviewer_JKCj · 2025-03-13

**Overall Recommendation:** 3

**Summary:**

The paper introduces Grammar Forced Translation (GraFT), a framework for translating natural language into temporal logic. GraFT simplifies the translation process by restricting the output tokens to a limited set, using the unique properties of each task and exploiting the known grammar of temporal logic during training and inference. This approach improves end-to-end translation accuracy and out-of-domain accuracy compared to state-of-the-art methods.

## update after rebuttal

I have no further comments and maintain the initial assessment of the paper.

**Claims And Evidence:**

Intuitively, the benefits of focusing on valid grammer states for a more focused gradient are readily apparent. The shortcomings of a Causal Language Modelling (CLM)-based approach in AP Grounding  with respect to a Masked Language Modeling (MLM)-based approach are sensible and results support the hypothesis.

In other literature, Temporal Logic (TL) describes a more general logic over time and subsumes Linear Temporal Logic (LTL). It is unclear how the logic operators defined in the Sec. 2.1 differ from standard LTL and the paper could replace all references to                               TL with LTL (or $\texttt{LTL}_f$).

**Essential References Not Discussed:**

References using Temporal Logic:

[1] Robust Counterexample-guided Optimization for Planning from Differentiable Temporal Logic, Dawson & Fan, IROS 2022

[2] Co-learning Planning and Control Policies Constrained by Differentiable Logic Specifications, Xiong et al, ICRA 2024

[3] Lang2LTL-2: Grounding Spatiotemporal Navigation Commands Using Large Language and Vision-Language Models, Liu et al, 2024

**Experimental Designs Or Analyses:**

Table 4 compares NL2TL with the proposed method (GraFT) which does not consider time intervals in the temporal logic. Are the comparisons fair given than NL2TL contains the added ability to handle these time intervals (as a part of Signal Temporal Logic)?

**Methods And Evaluation Criteria:**

The datasets used are off-the-shelf and provided from previous work. Comparisons made for performance as well as performance on training with a limited subset of the data are reasonable and provide suitable evidence that GraFT is more data efficient and robust than other CLM-based models.

**Other Comments Or Suggestions:**

- The double quotes should be fixed for LaTeX e.g. P2L67C2 “go to the red room → Use `` instead of "
- Minor typos:
    - P6L287C1 - “We know turn”

**Other Strengths And Weaknesses:**

N/A

**Questions For Authors:**

1. In Algorithm 1, how is the temporal logic grammar function (`grammar.get_valid()`) defined or implemented?
2. Can the approach be extended to handle time intervals as in NL2TL?

**Relation To Broader Scientific Literature:**

The proposed method makes advances in generating Temporal Logic specs from Natural Language which can be used as a step in various pipelines that require a formal language as input [1, 2]. A high accuracy in this translation could generalize various methods to accepting natural language as input (and not just an expression in formal logic). The method does not consider grounding predicates in other modalities like vision/images [3].

**Theoretical Claims:**

The theory provides an intuition to the effectiveness of grammar constraints and appears valid.

---

> ### Author Rebuttal · Authors · 2025-04-01
>
> **Remarks for all reviewers:** We would like to thank all reviewers for their time and detailed feedback. With two weak accepts and two weak rejects (with one weak reject indicating that they would consider raising their score if provided additional experimental results), the paper is on the borderline to be accepted. In this overall response, we summarize the strengths of the paper and how we have addressed the reviewers' concerns.
>
> 1. **Writing and organization**: Clear and well-structured writing with easy-to-follow organization [72U5]
> 2. **Ease of implementation**: The framework is conceptually simple yet effective, making it likely to be adopted by others [n686]
> 3. **Advantage of Encoder only vs. CLM**: Notable achievement in outperforming GPT-4o using only a BERT+T5 combination [72U5, JKCj]
> 4. **Limited number of APs during grounding** [VmqZ]: The AP grounding with BERT used a maximum of 5 APs per input sequence. To address this, we conducted an additional evaluation of BERT and GPT-4 on the AP grounding task (presented in our response to reviewer 72U5). Our new evaluation shows results on sequences with 6-10 APs and 11-15 APs, in addition to the original 1-5 AP examples used in the original evaluation. We believe that the original benchmarks only contained up to 5 APs, as humans typically struggle to reason about a larger number of entities simultaneously.
> 5. **Limited number of CLMs are used in the evaluation** [n686]. Our original  experimental evaluation only compared with one state-of-the-art CLM, GPT-4o-mini. To address this limitation, we performed multiple additional evaluations. In particular, we conducted two evaluations of AP grounding using Gpt-4 and GPT-4o. In addition, we conducted an evaluation of end-to-end translation with NL2TL using Gpt-4 grounded inputs. The results of these evaluations are presented in our response to reviewer n686.
> 6. **How does our work distinguish itself from existing work in the program synthesis community** [VmqZ, n686 ]: To the best of our knowledge, this is the first time that several of the concepts in our submission have been applied to NL-to-TL translation, which is a contribution itself. Moreover, we do not simply present an off the shelf implementation of existing techniques (i.e grammar constrained decoding). We contribute a novel approach to training seq2seq models for the NL-to-TL task, outlined in section 3.2. Lines 278-288 of our submission describes a key difference in our training approach when compared against existing grammar-based techniques. Additionally, the use of BERT for AP grounding appears to be an entirely novel application that yields significant benefits over the standard CLM-based approach.
>
> **Beginning of direct response to JKCj**:
>
> Q4: In Algorithm 1, how is the temporal logic grammar function (grammar.get_valid()) defined or implemented?
>
> A4: The grammar.get\_valid(state) function takes the current parser state for the sequence, which is maintained per sequence and is updated when each new token is processed using the grammar.update\_state(state, token\_id) function, where state is the current state and token_id is the token that we've just parsed. The grammar state holds a list of tasks that need to be completed before parsing ends. When a "(" token is parsed, we push a "(" element to the state stack for the sequence. For example, the state of a sequence may be ["formula", "(", "prop"] and the token we are parsing is "\_". When we call grammar.update\_state(state, "\_"), the new state is ["formula", "(", "prop\_"]. Now we must parse the next token. We will call grammar.get\_valid(state), which will return a set of valid token ID's {1, 2, 3, 4, 5}, because we know that "prop_" can only be followed by a digit 1-5 (based on our dataset; if we expected more props we would have increased this to include all potential prop IDs). The full implementation of our grammar is provided in the reproducibility zip file in /TIME/TemporalLogicGrammar.py.
>
> Q5: Can the approach be extended to handle time intervals as in NL2TL?
>
> A5:   Yes, this is absolutely possible. The only requirement for extending GraFT to other temporal logic formalism is the adaptation of the chosen formalism from terminal-based into token-based. We decided to limit the considered temporal logics to LTL in order to keep the paper more streamlined. However, we will consider extensions to time-bounded LTL and STL in our future work.
>
> Additionally, we will address the minor typos pointed out in the comments and suggestions.

---

### Official Review · Reviewer_n686 · 2025-03-13

**Overall Recommendation:** 3

**Summary:**

This paper introduces Grammar-Forced Translation (GraFT), a framework to translate natural language (NL) into temporal logic (TL) using large language models (LLMs).

**Claims And Evidence:**

The claims regarding the performance improvements and complexity reduction of GraFT are convincingly supported by empirical evidence.

**Essential References Not Discussed:**

n/a

**Experimental Designs Or Analyses:**

I think some implementation details (e.g., hyperparameter selection, number of runs per experiment) could be better documented to ensure reproducibility.

**Methods And Evaluation Criteria:**

Seems to make sense

**Other Comments Or Suggestions:**

The figures could be improved for better readability, particularly Figures 3 and 4

**Other Strengths And Weaknesses:**

1. The framework is conceptually simple yet effective, making it likely to be adopted by others
2. The approach is data-efficient, performing well even with limited training examples

3.The focus is primarily on improving accuracy rather than computational efficiency; it's unclear if GraFT introduces any additional computational overhead
4.While the approach reduces the need for domain-specific training data, it still requires a modest amount of in-domain data for optimal performance


I would like to see more empirical result with different LLM other than gpt4-o-mini. If the author can show this, I will raise my score.

**Questions For Authors:**

How does it differ from top-k sampling?

**Relation To Broader Scientific Literature:**

The paper builds upon and extends prior work in NL-to-TL translation in several ways. The work connects to broader research on structured prediction, grammar-guided generation, and efficient fine-tuning of language models.

**Theoretical Claims:**

I check the section 3.2.2, and it seems to make sense

---

> ### Author Rebuttal · Authors · 2025-04-01
>
> **Before proceeding, please read our response addressed to all reviewers found in our rebuttal for reviewer JKCj.**
>
> Q1: I think some implementation details (e.g., hyperparameter selection, number of runs per experiment) could be better documented to ensure reproducibility.
>
> A1: We have included training scripts for the models that should allow for our results to be reproduced. This was provided in the supplementary materials zip file. For the sake of clarity, we are happy to provide the exact hyperparameters used for training. BERT and T5 were both trained for 3 epochs with an LR of 2e-5. T5 was given |training data|/3 * 0.075 warm up steps. Identical hyperparameters were used for training T5 used in NL2TL and GraFT.
>
> Q2: I would like to see more empirical results with different LLM other than gpt4-o-mini. If the author can show this, I will raise my score.
>
> A2: We have performed an evaluation of NL2TL with GPT4o and GPT-4 as the grounding model, an end-to-end evaluation that uses GPT-4 as the grounding model, and an end-to-end evaluation of an T5 without an AP grounding model. These new results are presented below:
>
> **End-to-End Evaluation of NL2TL with more models (Updated with 4 new entries: ungrounded seq2seq trained on 500 and 2000 examples, NL2TL with GPT-4 each trained on 500 and 2000 examples):**
>
> | Approach           | Data Quantity | AP Grounding Model | Translation Model | CW    | GLTL  | Navi  |
> |--------------------|---------------|--------------------|-------------------|-------|-------|-------|
> | Ungrounded Seq2Seq | 500           | -                  | T5           | 59.60 | 46.80 | 43.40 |
> | NL2TL              | 500 | GPT-4o-mini | T5                              | 93.00 | 83.80 | 80.40 |
> | NL2TL              | 500           | GPT-4              | T5                | 91.50 | 82.60 | 83.70 |
> | GraFT              | 500           | BERT               | T5                | 97.70 | 91.50 | 85.00 |
> | Ungrounded Seq2Seq | 2000          | -                  | T5        | 68.10 | 55.90 | 56.30 |
> | NL2TL 	   | 2000              | GPT-4o-mini        | T5        | 98.20 | 97.40 | 86.70 |
> | NL2TL              | 2000          | GPT-4              | T5                | 96.70 | 96.20 | 90.00 |
> | GraFT              | 2000          | BERT               | T5                | 99.90 | 99.80 | 99.10 |
>
>
> Q3:How does it differ from top-k sampling?
>
> A3: In top-k sampling, we direct generation using the K highest scoring the tokens at each step. In Grammar-forced Training (GraFT), we first outright eliminate all of the grammatically incorrect tokens (based on the previous LABEL token, not previous PREDICTED token) before computing the loss. Likewise during inference with GraFT, we perform this same operation, but based on the previous predicted token, because of course labels are not available during evaluation. In top-k sampling, the model may still be informed by a list of tokens that includes grammatically invalid options. We eliminate that possibility altogether using grammar-forcing.
>
> Q4: The focus is primarily on improving accuracy rather than computational efficiency; it's unclear if GraFT introduces any additional computational overhead
>
> A4: Experimentally, we have observed that GraFT training requires 15-20% more time than ordinary training with T5. We will provide a figure that displays the computational overhead for GraFT vs T5 during training and inference, to be placed in the appendix of our final submission.
>
> Q6:The figures could be improved for better readability, particularly Figures 3 and 4
>
> A6: We can improve the readability by increasing the font size and reducing the presence of less-relevant details of the T5 architecture. These changes should bring the relevant details of our framework into focus. We will address these concerns in the final version of the paper- but we are unable to make changes to our submission PDF during the review period.

---

### Official Review · Reviewer_VmqZ · 2025-03-14

**Overall Recommendation:** 3

**Summary:**

The authors propose GraFT, an innovative framework that employs masks to ensure the syntax correctness of generated LTL programs. The approach first utilizes BERT to extract atomic propositions (APs) and map them into a predefined set of equivalent classes for co-references. Then, it leverages T5 to learn the translation step. GraFT improves end-to-end translation accuracy by 5.49% and out-of-domain translation accuracy by 14.06% on average across three benchmarks.

**Claims And Evidence:**

The authors list four contributions; however, they primarily describe the framework rather than explicitly stating the key contributions.

**Essential References Not Discussed:**

[1] Bunel, Rudy, et al. "Leveraging grammar and reinforcement learning for neural program synthesis." arXiv preprint arXiv:1805.04276 (2018).

[2] Netz, Lukas, Jan Reimer, and Bernhard Rumpe. "Using grammar masking to ensure syntactic validity in llm-based modeling tasks." Proceedings of the ACM/IEEE 27th International Conference on Model Driven Engineering Languages and Systems. 2024.

**Experimental Designs Or Analyses:**

See Methods And Evaluation Criteria

**Methods And Evaluation Criteria:**

I have several questions regarding the methodology:

**Q1.** Why is the vocabulary size for atomic propositions (APs) limited to only 6? If the narration spans an entire movie lasting over two hours, wouldn’t it require significantly more atomic propositions?

**Q2.** How are the grounded nouns utilized in the **LTL** formula? Could you provide an example?

**Q3.** The dataset descriptions are insufficiently explained. Regarding the **GW** dataset, I could not find details on its context, size, input, and output in either the main text or the supplementary material.

**Other Comments Or Suggestions:**

For data efficiency and accuracy results, it would be nice to compare against the NL2TL baseline.

**Other Strengths And Weaknesses:**

My main concern with this work is how it differentiates itself from existing literature in the program synthesis community.

**Questions For Authors:**

See Methods And Evaluation Criteria, and Other Strengths And Weaknesses

**Relation To Broader Scientific Literature:**

This work is highly relevant to the syntax-guided program synthesis community. In fact, the masking strategy used to ensure program syntax correctness has been around for quite some time. See Essential References Not Discussed for further reference.

**Theoretical Claims:**

I have check the algorithm and it makes sense.

---

> ### Author Rebuttal · Authors · 2025-04-01
>
> **Before proceeding, please read our response addressed to all reviewers found in our rebuttal for reviewer JKCj.**
>
> Q1. Why is the vocabulary size for atomic propositions (APs) limited to only 6? If the narration spans an entire movie lasting over two hours, wouldn’t it require significantly more atomic propositions?
>
> A1. This limitation was introduced by the existing datasets rather than by the authors. We choose a prop id space of 6 tokens because the sentences in our dataset have a maximum of 5 APs. Each token that is not part of these is assigned the 0-class, and any tokens which are part of an AP are assigned an integer ID corresponding to which AP it is a part of. However, our proposed approach can certainly be extended to handle more than 5 APs. **Please see our response to reviewer 72U5, where we have provided results for up to 15 APs.** For any number of APs, our approach of performing grounding using an encoder only model outperforms using CLMs.
>
> We would also like to point out that in the “2 hour long narration” example, it seems that the reviewer is referring to a *trace* with a large number of AP's, rather than a temporal logic expression. The difference being a *trace* is a time-ordered sequence of states over which some specification can be checked (the specification being a temporal logic expression). For example, we may specify in natural language that “The identity of the culprit is not revealed until the film is in the third act.”, and we may check that specification against a trace (i.e, extended document that holds the dialogue). Moreover, we believe that the 5 AP maximum observed in the dataset reflects the general expectation that the input NL sentences to our framework are roughly equivalent to a single human query, which typically do not contain hundreds of APs.
>
> Q2: How are the grounded nouns utilized in the LTL formula? Could you provide an example?
>
> A2: The grounded AP's represent the parts of the sentence which will serve as atomic predicates in the resulting TL expression. An example NL input may be: "When the apple falls from the tree, pick it up and eventually put it in the basket." The grounding system would label every token in the input with the AP it relates to. For the purposes of this example I will just label the words rather than the true subword tokens used by BERT. The output sequence from BERT may look like: [0,1,1,1,1,1,1,0,2,2,2,0,0,3,3,3,3,3], and we construct a new sentence based on this sequence: "When the prop_1, prop_2 and eventually prop_3." And we store the corresponding tokens in a dictionary: {prop_1: "the apple falls from the tree", prop_2: "pick it up", prop_3: "put it in the basket"}. The purpose of this is to simplify the NL prior to translation without sacrificing altogether the semantic nuance and complexity of the original natural language; the relevant segments are stored in a dictionary so they can be used downstream to determine their truth values.
>
> Q3: The dataset descriptions are insufficiently explained. Regarding the GW dataset, I could not find details on its context, size, input, and output in either the main text or the supplementary material.
>
> A3: We do not reference any GW dataset - Is this question perhaps about CW or GLTL? If so, there is a table in the appendix (A1) with information on the datasets. The table contains the number of unique NL sentences, number of unique LTL sentences, and the number of unique words that appear in each dataset.
>
> Q4: For data efficiency and accuracy results, it would be nice to compare against the NL2TL baseline.
>
> A4: We have performed the requested evaluation and the results can be viewed using the link (https://docs.google.com/document/d/e/2PACX-1vTYbY_0G_5EUXs1sumgRB7q3zz_3gCQRU0g8OCFXGFvxH2thZ2NqFNzxXWujADzOTj1uBICDQxRCcFi/pub). The results show that our approach outperforms NL2TL with even more margin than the T5 baseline. This stems from the fact that our method of grounding the APs is superior to the method in NL2TL. We had originally used T5 to isolate the improvement from the training.
>
> In regard to the missing essential reference: This does appear to be an essential reference which we should include in our discussion of the background work. While the authors approach to pruning the space of target programs appears similar to our proposed approach (GraFT), there is a key difference relating to how we obtain the grammatical state (or as the authors describe it on page 7, the “current context”). In the author’s approach, the syntax checker functions the same during both training and inference, establishing the current context using previously generated tokens. However in GraFT, the syntactic validity of tokens at position t is dependent on the label at position t-1, rather than the generated token at position t-1, as would be used in the author’s proposed method.

---

> > ### Comment · Reviewer_VmqZ · 2025-04-03
> >
> > Thank you for clarifying my questions. I have raised my score accordingly.

---

> > > ### Author Response · Authors · 2025-04-08
> > >
> > > We thank all reviewers for their careful reading of our submission, and for providing valuable feedback. Our work was recognized for its clear organization, ease of implementation, and promising results in natural language to temporal logic translation. While the reviewers initial evaluation considered the paper borderline (two weak accepts, two weak rejects), we resolved the reviewers' concerns during the rebuttal period and there is now a consensus among the reviewers to accept the paper (four weak accepts).
> > >
> > > Key Strengths
> > >
> > > Clarity, organization, and reproducibility: Multiple reviewers commended the paper for clear writing and logical structure. We maintain this clarity in our explanation of additional evaluations. Additionally, we have supplied the code required to reproduce our results in the supplemental materials, as well as a description of our training environment and hyperparameters.
> > >
> > > Encoder-only approach: One of our notable contributions is outperforming strong generative models (including GPT-4o) using BERT. This approach demonstrates the unique advantages of masked language models that are often overlooked in recent work.
> > >
> > > Practical value: The GraFT framework is intuitive and straightforward to implement, and integrates well with other seq2seq models, making it likely to be adopted by the community.
> > >
> > > Novelty: Our submission is, to our knowledge, the first to apply BERT to the task of AP grounding. Additionally, the limited prior work on grammar-constrained training maintains a grammatical context with respect to previously generated tokens (as would be done in grammar-constrained decoding). In contrast, our approach leverages the insight that such sequences are rarely correct in the early stages of training, leading us to force grammatical constraints relative to the ground-truth sequence, rather than the (often erroneous) sequence generated during training.
> > >
> > > Resolved Concerns:
> > > Limited number of APs: We supply an additional evaluation that demonstrates the continued success of our BERT-based AP grounding approach for AP quantities up to 15.
> > >
> > > Additional LLMs: We conducted an additional evaluation in which GPT-4 and GPT-4o were used for grounding and translation, in order to fully explore the current capabilities of generative models.
> > >
> > > End-to-end Evaluation: We supply additional entries in our end-to-end evaluation, including an evaluation of unaided language models, and NL2TL with different foundation models functioning as the AP masker.
> > >
> > > By providing more extensive evaluations, clarifying our methodology, and comparing against additional baselines, we have resolved the primary issues raised by the reviewers. We believe our paper offers a comprehensive contribution: it introduces a novel and effective method for NL-to-TL translation, clearly justifies its design choices, and offers robust empirical evidence of its advantages across various scenarios. We again thank all reviewers for their insightful comments and believe that our revisions substantively address their feedback, as evidenced by their updated evaluations.

---

### Official Review · Reviewer_72U5 · 2025-03-14

**Overall Recommendation:** 3

**Summary:**

This paper introduces a framework for translating natural language to temporal logic by restricting output token space during both grounding and translation phases. The framework employs a masked language model for atomic propositions grounding and a fine-tuned sequence-to-sequence model for translation. Using a BERT+T5 combination, GraFT demonstrates significant improvements in both end-to-end translation accuracy and out-of-domain scenarios compared to existing methods. The paper provides mathematical justification for token restriction benefits and evaluates the framework on CW, GLTL, and Navigation benchmarks.

**Claims And Evidence:**

Yes.

**Essential References Not Discussed:**

No.

**Experimental Designs Or Analyses:**

Yes. See **Weaknesses** and **Strengths** below.

**Methods And Evaluation Criteria:**

Yes.

**Other Comments Or Suggestions:**

N/A

**Other Strengths And Weaknesses:**

**Weaknesses:**
- The method for obtaining grounded NL sequences for T5 training lacks clear explanation. The paper fails to justify whether these grounded NL sequences are suitable as learning objectives for CLMs.
- The difference in output formats between CLM and MLM for grounded NL (Figure 2) requires clarification - the paper should explain whether this stems from fundamental architectural differences between the two models



**Strengths:**
- Clear and well-structured writing with easy-to-follow organization
- Innovation points are well-articulated
- Results are presented clearly and comprehensively
- Notable achievement in outperforming GPT-4o using only a BERT+T5 combination

**Questions For Authors:**

**Questions:**

- The baseline models used for comparison (from 2023) may be outdated. More recent baselines should be considered to strengthen the comparative analysis

**Relation To Broader Scientific Literature:**

GraFT's token space restriction approach bridges general NLP techniques with formal language translation tasks.

**Theoretical Claims:**

No.

---

> ### Author Rebuttal · Authors · 2025-04-01
>
> **Before proceeding, please read our response addressed to all reviewers found in our rebuttal for reviewer JKCj.**
>
> Q1: The method for obtaining grounded NL sequences for T5 training lacks clear explanation. The paper fails to justify whether these grounded NL sequences are suitable as learning objectives for CLMs.
>
> A1: In GraFT, the grounded sentences are obtained by passing the original NL sentences to BERT, which provides a mapping between the AP labels and segments of the sentence. The BERT model must be fine-tuned on labeled data, which is provided in the LTL datasets (these datasets include information on the span of APs in the NL sentences). In the NL2TL framework, an LLM is prompted to generate the AP labels and segments within the sentence, rather than directly labeling the tokens of the input as is done by BERT. In response to the question of whether grounding NL sequences is suitable as a learning objective for CLMs, we present the tables below. The tables contain AP grounding results for 3 ranges of AP quantities (1-5 (A), 6-10 (B), and 11-15 (C)) for an MLM and an off-the-shelf CLM. Our position is that CLMs are *not* an optimal choice for this learning objective.
>
> (A)
> | Model       | Objective | CW (%) | GLTL (%) | Navi (%) |
> |-------------|-----------|--------|----------|----------|
> | GPT-4o-mini | Causal    | 97.76  | 95.84    | 83.97    |
> | GPT-4o      | Causal    | 95.02  | 93.53    | 86.08    |
> | GPT-4       | Causal    | 96.24  | 94.68    | 87.28    |
> | DistilBERT  | Masked    | 95.80  | 93.83    | 99.99    |
> | RoBERTa     | Masked    | 98.34|  96.96     | 99.99|
> | BERT 	| Masked | 98.58  | 97.35    | 99.99    |
> (B)
>
> | Model      | Objective | CW (%) | GLTL (%) | Navi (%) |
> |-------------|-----------|--------|----------|----------|
> | GPT-4      | Causal    | 81.88  | 80.41    | 83.02    |
> | DistilBERT | Masked    | 94.20  | 91.75    | 99.99    |
> | RoBERTa    | Masked    | 96.37  | 95.66    | 99.99    |
> | BERT       | Masked    | 97.10  | 96.52    | 99.99    |
> (C)
>
> | Model      | Objective | CW (%) | GLTL (%) | Navi (%) |
> |-------------|-----------|--------|----------|----------|
> | GPT-4      | Causal    | 70.34  | 69.80    | 72.24    |
> | DistilBERT | Masked    | 92.86  | 90.63    | 98.54    |
> | RoBERTa    | Masked    | 95.13  | 94.67    | 99.73    |
> | BERT       | Masked    | 95.78  | 96.44    | 99.91    |
>
>
> Q2: The difference in output formats between CLM and MLM for grounded NL (Figure 2) requires clarification - the paper should explain whether this stems from fundamental architectural differences between the two models
>
> A2: The fundamental difference is that BERT classifies each token in the input as PROP_ID or 0, where the PROP_ID class is an integer that corresponds to an AP that appears one or more times in the sentence, and the 0 class indicates that a token is not part of any AP. In contrast, a CLM is *generative* and predicts (the next) future token, rather than classifying the tokens it received as input. Our evaluation results for MLM vs CLM AP grounding (provided above) support our position that the MLM training objective is better suited to the task of AP grounding than the CLM objective.
>
> Q3: The baseline models used for comparison (from 2023) may be outdated. More recent baselines should be considered to strengthen the comparative analysis
>
> A3: In our supplemental evaluation results provided in our response to reviewer **n686**, we include recently trained models including GPT-4 and GPT-4o, trained in 2024. To our knowledge, our evaluation compares against the current state-of-the-art temporal logic translation frameworks.

---

### Decision · Program_Chairs · 2025-05-01

**Decision:**

Accept (poster)

**Comment:**

The paper proposes a framework for converting natural language to temporal logic specifications (such as LTL). This is achieved by restricting the output token space during grounding (where a masked language model is used) and a during translation (using a fine-tuned seq-to-seq model). Results demonstrate that the framework - a combination of BERT + T5 - outperforms existing baselines, including LLMs.

Reviewers were generally positive about the paper, which provided theoretical intuition and a convincing set of experiments, outperforming more modern LLMs. The authors provided clear responses in their rebuttal, including clarification of finer details, context for the work within the program synthesis space, and additional experiments comparing to other large language models. These additions result in a stronger paper that is highly relevant to the community, and could see wide adoption due to its relative simplicity and excellent performance.